# AN INTERPRETABLE GRAPH GENERATIVE MODEL WITH HETEROPHILY

## ABSTRACT

Many models for graphs fall under the framework of edge-independent dot product models. These models output the probabilities of edges existing between all pairs of nodes, and the probability of a link between two nodes increases with the dot product of vectors associated with the nodes. Recent work has shown that these models are unable to capture key structures in real-world graphs, particularly heterophilous structures, wherein links occur between dissimilar nodes. We propose the first edge-independent graph generative model that is a) expressive enough to capture heterophily, b) produces nonnegative embeddings, which allow link predictions to be interpreted in terms of communities, and c) optimizes effectively on real-world graphs with gradient descent on a cross-entropy loss. Our theoretical results demonstrate the expressiveness of our model in its ability to exactly reconstruct a graph using a number of clusters that is linear in the maximum degree, along with its ability to capture both heterophily and homophily in the data. Further, our experiments demonstrate the effectiveness of our model for a variety of important application tasks such as multi-label clustering and link prediction.

## 1 INTRODUCTION

Graphs naturally arise in data from a variety of fields including sociology (Mason & Verwoerd, 2007), biology (Scott, 1988), and computer networking (Bonato, 2004). A key underlying task in machine learning for graph data is forming models of graphs which can predict edges between nodes, form useful representations of nodes, and reveal interpretable structure in the graph, such as detecting clusters of nodes. Many graph models fall under the framework of edge-independent graph generative models, which can output the probabilities of edges existing between any pair of nodes. The parameters of such models can be trained iteratively on the network, or some fraction of the network which is known, in the link prediction task, e.g., by minimizing a cross-entropy loss. To choose among these models, one must consider whether the model is capable of expressing structures of interest in the graph, as well as the interpretability of the model.

**Expressiveness** As real-world graphs are high-dimensional objects, graph models generally compress information about the graph. Such models are exemplified by the family of dot product models, which associate each node with a real-valued "embedding" vector; the predicted probability of the link between two nodes increases with the dot product of their embedding vectors. These models can alternatively be seen as factorizing the adjacency matrix of the graph in terms of a low-rank matrix. Recent work (Seshadhri et al., 2020) has shown that dot product models are limited in their ability to model common structures in real-world graphs, such as triangles incident only on low-degree nodes. In response, Chanpuriya et al. (2020) showed that with the logistic PCA (LPCA) model, which has two embeddings per node (i.e. using the dot product of the "left" embedding of one node and the "right" embedding of another), not only can such structures be represented, but further, any graph can be exactly represented with embedding vectors whose lengths are linear in the maximum degree of the graph. Peysakhovich & Bottou (2021) show that the limitations of the single-embedding model, which are overcome by having two embeddings, stem from only being able to represent adjacency matrices which are positive semi-definite, which prevents them from representing heterophilous structures in graphs; heterophilous structures are those wherein dissimilar nodes are linked.

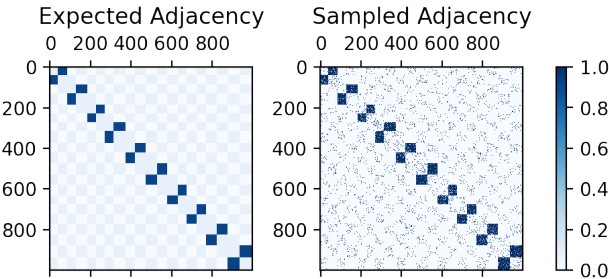

Figure 1: The motivating synthetic graph. The expected adjacency matrix (left) and the sampled matrix (right); the latter is passed to the training algorithms. The network is approximately a union of ten bipartite graphs, each of which correspond to recruiters and non-recruiters at one of the ten locations.

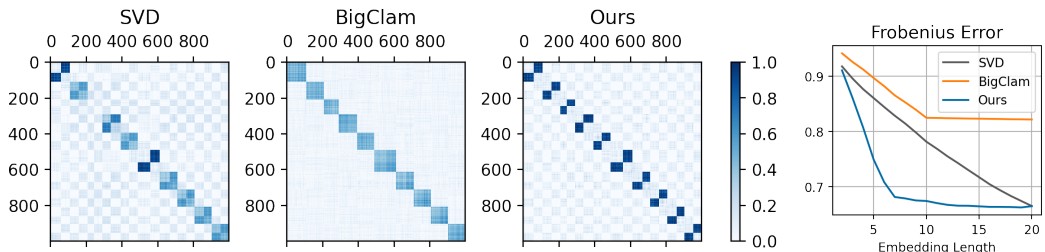

Figure 2: (Right) Reconstructions of the motivating synthetic graph of Figure 1 with SVD, BIGCLAM, and our model, using 12 communities or singular vectors. Note the lack of the small diagonal structure in BIGCLAM's reconstruction; this corresponds to its inability to capture the heterophilous interaction between recruiters and non-recruiters. (Left) Frobenius error when reconstructing the motivating synthetic graph of Figure 1 with SVD, BIGCLAM, and our model, as the embedding length is varied. The error is normalized by the sum of the true adjacency matrix (i.e., the number of edges).

**Heterophily: Motivating example**    To demonstrate how heterophily can manifest in networks, as well as how models which assume homophily can fail to represent such networks, we provide a simple synthetic example. Suppose a recruiting website allows its members to contact each other; we construct a graph of these members, with an edge indicating that two members have been in contact. Members are either recruiters or non-recruiters, and each member comes from one of ten locations (e.g., a city). Members from the same location are likely to contact each other; this typifies homophily, wherein links occur between similar nodes. Furthermore, recruiters are unlikely to contact other recruiters, and non-recruiters are unlikely to contact other non-recruiters; this typifies heterophily. Figure 1 shows an instantiation of such an adjacency matrix with 1000 nodes, which are randomly assigned to one of the ten locations and one of recruiter / non-recruiter. We recreate this network with our embedding model and the BIGCLAM algorithm of Yang & Leskovec (2013), which explicitly assumes homophily. We also compare with the best low-rank approximation to the adjacency matrix in terms of Frobenius error; this is the SVD of the matrix, discarding all but the top singular values. In Figure 1, we show how BIGCLAM captures only the ten communities based on location, i.e., only the homophilous structure, and fails to capture the heterophilous distinction between recruiters and non-recruiters. We also plot the error of the reconstructions as the embedding length increases. There are $10 \cdot 2 = 20$ different kinds of nodes, meaning the expected adjacency matrix is rank-20, and our model maintains the lowest error up to this embedding length; by contrast, BIGCLAM is unable to decrease error after capturing location information with length-10 embeddings.

**Interpretability**    Beyond being able to capture a given network accurately, it is often desirable for a graph model to form interpretable representations of nodes and to produce edge probabilities in an interpretable fashion. Dot product models can achieve this by restricting the node embeddings to be nonnegative. Nonnegative factorization has long been used to decompose data into parts (Donoho & Stodden, 2003). In the context of graphs, this entails decomposing the set of nodes of the network into clusters or communities. In particular, each entry of the nonnegative embedding vector of a node

represents the intensity with which the node participates in a community. Note that this allows the edge probabilities output by dot product models to be interpretable in terms of coparticipation in communities. Depending on the model, these vectors may have restrictions such as a sum-to-one requirement, meaning the node is assigned a categorical distribution over communities. The least restrictive and most expressive case is that of soft assignments to overlapping communities, where the entries can vary totally independently. In models for this case, the output of the dot product is often mapped through a nonlinear link function to produce a probability, i.e. to ensure the value lies in $[0, 1]$. This link function ideally also facilitates straightforward interpretation.

We propose the first edge-independent graph generative model that is a) expressive enough to capture heterophily, b) interpretable in that it produces nonnegative embeddings, and c) optimizes effectively on real-world graphs with gradient descent on a cross-entropy loss.

**Summary of main contributions**   The key contributions of this work are as follows:

- We introduce a graph generative model, based on nonnegative matrix factorization, which is able to represent both heterophily and overlapping communities. Our model outputs link probabilities which are interpretable in terms of the communities it detects.

- We provide a scheme for initialization of the nonnegative factors using the arbitrary real factors generated by logistic PCA. We show theoretically how a graph which is represented exactly by LPCA can also be represented exactly by our model.

- We show theoretically that, with a small number of communities, our model can exactly represent a natural class of graphs which exhibits both heterophily and overlapping communities.

- In experiments, we show that our algorithm is competitive on real-world graphs in terms of representing the network, doing link prediction, and producing communities which align with ground-truth.

## 2   GRAPH GENERATIVE MODEL

Consider the set of undirected, unweighted graphs on $n$ nodes, i.e., the set of graphs with symmetric adjacency matrices in $\{0, 1\}^{n \times n}$. We propose an edge-independent, generative model for such graphs. Given a diagonal matrix $\boldsymbol{W} \in \mathbb{R}^{k \times k}$ and a matrix $\boldsymbol{V} \in [0, 1]^{n \times k}$, we set the probability of an edge existing between nodes $i$ and $j$ to be the $(i, j)$-th entry of matrix $\tilde{\boldsymbol{A}}$:

$$\tilde{\boldsymbol{A}} := \sigma(\boldsymbol{V}^\top \boldsymbol{W} \boldsymbol{V}), \tag{1}$$

where $\sigma$ is the logistic function. $k$ represents the number of clusters; intuitively, if $\mathbf{v}_i \in \mathbb{R}^k$ is the $i$-th row of matrix $\boldsymbol{V}$, then $\mathbf{v}_i$ is soft assignment of node $i$ to the $k$ communities. $\boldsymbol{W}$ can be viewed as a cluster affinity matrix. An equivalent alternative formulation is

$$\tilde{\boldsymbol{A}}_{i,j} = \sigma(\mathbf{v}_i \boldsymbol{W} \mathbf{v}_j^\top). \tag{2}$$

**Interpretation**   The edge probabilities output by this model have an intuitive interpretation, and to maximize interpretability, we focus on the case where $\boldsymbol{W}$ is diagonal. Recall that there is a one-to-one-to-one relationship between probability $p \in [0, 1]$, odds $o = \frac{p}{1-p} \in [0, \infty)$, and logit $\ell = \log(o) \in (-\infty, +\infty)$. The logit of the link probability between nodes $i$ and $j$ is $\mathbf{v}_i^\top \boldsymbol{W} \mathbf{v}_j$, which is a summation of terms $\mathbf{v}_{ic} \mathbf{v}_{jc} \boldsymbol{W}_{cc}$ over all communities $c \in [k]$. If the nodes both fully participate in community $c$, that is, $\mathbf{v}_{ic} = \mathbf{v}_{jc} = 1$, then the edge logit is changed by $\boldsymbol{W}_{cc}$ starting from a baseline of 0, or equivalently the odds of an edge is multiplied by $\exp(\boldsymbol{W}_{cc})$ starting from a baseline odds of 1; if either of the nodes participates only partially in community $c$, then the change in logit and odds is accordingly prorated. Homophily and heterophily also have a clear interpretaion in this model: homophilous communities are those with $\boldsymbol{W}_{cc} > 0$, where two nodes both participating in the community increases the odds of a link, whereas communities with $\boldsymbol{W}_{cc} < 0$ are heterophilous, and coparticipation decreases the odds of a link.

## 3   RELATED WORK

**Node clustering**   There is extensive prior work on the node clustering problem (Schaeffer, 2007; Aggarwal & Wang, 2010; Nascimento & De Carvalho, 2011), perhaps the most well-known being the normalized cuts algorithm of Shi & Malik (2000), which produces a clustering based on the entrywise signs of an eigenvector of the graph Laplacian matrix. However, the clustering algorithms which are most relevant to our work are those based on non-negative matrix factorization (NMF) (Lee & Seung, 1999; Berry et al., 2007; Wang & Zhang, 2012; Gillis, 2020). One such algorithm is that of Yu et al. (2005), which approximately factors a graph's adjacency matrix $A \in \{0,1\}^{n \times n}$ into two positive matrices $H$ and $\Lambda$, where $H \in \mathbb{R}_+^{n \times k}$ is left-stochastic (i.e. each of its columns sums to 1) and $\Lambda \in \mathbb{R}_+^{k \times k}$ is diagonal, such that $H \Lambda H^\top \approx A$. Here $H$ represents a soft clustering of the $n$ nodes into $k$ clusters, while the diagonal entries of $\Lambda$ represent the prevalence of edges within clusters. Note the similarity of the factorization to our model, save for the lack of a nonlinearity. Other NMF approaches include those of Ding et al. (2008), Yang et al. (2012), Kuang et al. (2012), and Kuang et al. (2015) (SYMNMF).

**Modeling heterophily**   Much of the existing work on graph models has an underlying assumption of network homophily (Newman, 2002; Johnson et al., 2010; Noldus & Van Mieghem, 2015). There has been significant recent interest in the limitations of graph neural network (GNN) models (Duvenaud et al., 2015; Li et al., 2016; Kipf & Welling, 2017; Hamilton et al., 2017) at addressing network heterophily (Nt & Maehara, 2019; Zhu et al., 2020), as well as proposed solutions (Pei et al., 2020; Zhu et al., 2021; Yan et al., 2021), but relatively less work for more fundamental models such as those for clustering. Some existing NMF approaches to clustering do naturally model heterophilous structure in networks. The model of Nourbakhsh et al. (2014), for example, is similar to that of Yu et al. (2005), but allows the cluster affinity matrix $\Lambda$ to be non-diagonal; this allows for inter-cluster edge affinity to exceed intra-cluster edge affinity, so heterophily can arise in this model, though it is not a focus of their work. Further, the model of Miller et al. (2009) is similar to ours and also allows for heterophily, though it restricts the cluster assignment matrix $V$ to be binary; additionally, their training algorithm is not based on gradient descent as ours is, and it does not scale to large networks. More recently, Peysakhovich & Bottou (2021) propose a decomposition of the form $A \approx D + BB^\top - CC^\top$, where $D \in \mathbb{R}^{n \times n}$ is diagonal and $B, C \in \mathbb{R}^{n \times k}$ are low-rank; the authors discuss how, interestingly, this model separates the homophilous and heterophilous structure into different factors, namely $B$ and $C$. However, this work does not pursue a clustering interpretation or investigate setting the factors $B$ and $C$ to be nonnegative. One stage of our training algorithm uses a similar decomposition, though it includes the nonnegativity constraint; this is detailed in Section 4.

**Overlapping clustering**   Many models discussed above focus on the single-label clustering task. We are interested in the closely-related but distinct task of multi-label clustering, also known as overlapping community detection (Xie et al., 2013; Javed et al., 2018). The BIGCLAM algorithm of Yang & Leskovec (2013) uses the following generative model for this task: the probability of a link between two nodes $i$ and $j$ is given by $1 - \exp(-\boldsymbol{f}_i \cdot \boldsymbol{f}_j)$, where $\boldsymbol{f}_i, \boldsymbol{f}_j \in \mathbb{R}_+^k$ represent the intensities with which the nodes participate in each of the $k$ communities. This model allows for intersections of communities to be especially dense with edges, which the authors generally observe in real-world networks; by contrast, they claim that prior state-of-the-art approaches, including ones based on clustering links (Ahn et al., 2010) and clique detection (Palla et al., 2005), as well as a mixed-membership variant (Airoldi et al., 2008) of the stochastic block model (Holland et al., 1983), implicitly assume that intersections are sparse. BIGCLAM assumes strict homophily of the communities, whereas our model allows for both homophily and heterophily. Additionally, unlike in our model, there is no upper bound to the intensities of community participation (i.e. the entries of each $\boldsymbol{f}$), so it is unclear how to incorporate prior knowledge about community membership in the form of binary labels, as in a semi-supervised situation.

The approach of Zhang & Yeung (2012) is more similar to ours and more amenable to such prior information in that community assignments are bounded; specifically, the model is similar to those of Yu et al. (2005) and Nourbakhsh et al. (2014), but allows the cluster assignment matrix $H$ to be an arbitrary matrix of probabilities rather left-stochastic. However, unlike our model and BIGCLAM, these models lack a nonlinear linking function; recent work outside clustering and community detection on graph generative models (Rendsburg et al., 2020; Chanpuriya et al., 2020) suggests that the addition of a nonlinear linking function, specifically softmax and logistic nonlinearities as in our

model, can make matrix factorization-based graph models more expressive. Lastly, a recent approach is the VGRAPH model of Sun et al. (2019), which also lacks a final nonlinear linking function, but, interestingly, has an intermediate linking function: the matrix factors (i.e. the cluster assignment matrices) themselves are a product of learned embeddings for the nodes and communities, put through a softmax linking function. Their algorithm ultimately determines overlapping communities as in link clustering approaches, and they find that it generally achieves state-of-the-art results in matching ground-truth communities; as discussed in Section 6.2, we find that our algorithm's performance on this task compares favorably to VGRAPH.

## 4 TRAINING ALGORITHM

Given an input graph $A \in \{0, 1\}^{n \times n}$, we find $V$ and $W$ such that the model produces $\tilde{A} = \sigma(VWV^\top) \in (0, 1)^{n \times n}$ as in Eq. (1) which approximately matches $A$. In particular, we train the model to minimize the sum of binary cross-entropies of the link predictions over all pairs of nodes:

$$R = -\sum \left( A \log(\tilde{A}) \right) - \sum \left( (1 - A) \log(1 - \tilde{A}) \right), \tag{3}$$

where $\sum$ denotes the scalar summation of all entries in the matrix. Rather than optimizing the model of Equation 1 directly, we optimize different parametrizations which we find are more effective. This optimization comprises three stages. Note that while we outline a non-stochastic version of the algorithm, each stage can generalize straightforwardly to a stochastic version, i.e., by sampling links and non-links for the loss function.

**First stage** We first fit the unconstrained logistic principal components analysis (LPCA) model to the input graph as in Chanpuriya et al. (2020). This model reconstructs a graph $\tilde{A} \in \{0, 1\}^{n \times n}$ using logit factors $X, Y \in \mathbb{R}^{n \times k}$ via the model

$$\tilde{A} = \sigma(XY^\top). \tag{4}$$

Factors $X$ and $Y$ are initialized randomly, then trained via gradient descent on the loss of Equation 3 so that $\tilde{A} \approx A$. Note that entries of the factors $X$ and $Y$ are not necessarily nonnegative; hence this model does not directly admit an interpretation as community detection. Unlike Chanpuriya et al. (2020), which explicitly seeks to exactly fit the graph, i.e., to find $X, Y$ such that $\tilde{A} = A$, and does not explore the graph structure which is recovered in the factors, we employ $L_2$ regularization of the factors to avoid overfitting. See Algorithm 1 for pseudocode of this stage.

**Second stage** The factors $X$ and $Y$ from the first stage are processed into nonnegative factors $B \in \mathbb{R}_+^{n \times k_B}$ and $C \in \mathbb{R}_+^{n \times k_C}$ such that $k_B + k_C = 3k$ and

$$BB^\top - CC^\top \approx \tfrac{1}{2} \left( XY^\top + YX^\top \right).$$

Note that the left-hand side can only represent symmetric matrices. Let $L = \tfrac{1}{2} \left( XY^\top + YX^\top \right)$. $L$ is a symmetrization of $XY^\top$; if $\sigma(XY^\top)$ closely approximates the symmetric matrix $A$ as desired, so too should the symmetrized logits. Pseudocode for this stage is given in Algorithm 2. The concept of this stage is to first separate the logit matrix $L$ into a sum and difference of rank-1 components via eigendecomposition. Each of these components can be written as $+\mathbf{v}\mathbf{v}^\top$ or $-\mathbf{v}\mathbf{v}^\top$ with $\mathbf{v} \in \mathbb{R}^n$, where the sign depends on the sign of the eigenvalue. Each component is then separated into a sum or difference of three outer products of nonnegative vectors, via the claim below.

**Claim 4.1.** *Let $\phi : \mathbb{R} \to \mathbb{R}$ denote the ReLU activation function, i.e., $\phi(z) = \max\{z, 0\}$. For any vector $\mathbf{v}$,*

$$\mathbf{v}\mathbf{v}^\top = 2\phi(\mathbf{v})\phi(\mathbf{v})^\top + 2\phi(-\mathbf{v})\phi(-\mathbf{v})^\top - |\mathbf{v}||\mathbf{v}|^\top$$

*Proof.* Take any $\mathbf{v} \in \mathbb{R}^k$. Then

$$\begin{aligned}
\mathbf{v}\mathbf{v}^\top &= (\phi(\mathbf{v}) - \phi(-\mathbf{v})) \cdot (\phi(\mathbf{v})^\top - \phi(-\mathbf{v})^\top) \\
&= \phi(\mathbf{v})\phi(\mathbf{v})^\top + \phi(-\mathbf{v})\phi(-\mathbf{v})^\top - \phi(\mathbf{v})\phi(-\mathbf{v})^\top - \phi(-\mathbf{v})\phi(\mathbf{v})^\top \\
&= 2\phi(\mathbf{v})\phi(\mathbf{v})^\top + 2\phi(-\mathbf{v})\phi(-\mathbf{v})^\top - (\phi(\mathbf{v}) + \phi(-\mathbf{v})) \cdot (\phi(\mathbf{v}) + \phi(-\mathbf{v}))^\top \\
&= 2\phi(\mathbf{v})\phi(\mathbf{v})^\top + 2\phi(-\mathbf{v})\phi(-\mathbf{v})^\top - |\mathbf{v}||\mathbf{v}|^\top,
\end{aligned}$$

where the first step follows from $\mathbf{v} = \phi(\mathbf{v}) - \phi(-\mathbf{v})$, and the last step follows from $|\mathbf{v}| = \phi(\mathbf{v}) + \phi(-\mathbf{v})$. ∎

Algorithm 2 constitutes a constructive proof of the following theorem.

**Theorem 4.2** (Nonnegative Factorization of Rank-$k$ Matrices). *Given a symmetric rank-$k$ matrix $\boldsymbol{L} \in \mathbb{R}^{n \times n}$, there exist nonnegative matrices $\boldsymbol{B} \in \mathbb{R}_+^{n \times k_B}$ and $\boldsymbol{C} \in \mathbb{R}_+^{n \times k_C}$ such that $k_B + k_C = 3k$ and $\boldsymbol{BB}^\top - \boldsymbol{CC}^\top = \boldsymbol{L}$.*

**Third stage** The factors $\boldsymbol{B}$ and $\boldsymbol{C}$ from the previous stage serve as initialization for the final stage of optimization. Of the $3k$ communities generated by the previous stage, we keep the top $k$ which are most impactful on the edge logits, as ranked by the $L_2$ norms of the columns of $\boldsymbol{B}$ and $\boldsymbol{C}$. Now $\boldsymbol{B} \in \mathbb{R}_+^{n \times k_B}$ and $\boldsymbol{C} \in \mathbb{R}_+^{n \times k_C}$ such that $k_B + k_C = k$.

These remaining $k$ communities are then directly optimized by minimizing the cross-entropy loss of Equation 3 on the following graph model:

$$\tilde{\boldsymbol{A}} = \sigma\left(\boldsymbol{BB}^\top - \boldsymbol{CC}^\top\right). \tag{5}$$

This stage proceeds exactly as the first stage, i.e. as in Algorithm 1, except with Equation 5 as the generative model rather than Equation 4. Additionally, the optimized parameters $\boldsymbol{B}$ and $\boldsymbol{C}$ are constrained to be nonnegative.

The model in Equation 5 is exactly equivalent to that of Equation 1, where $\boldsymbol{W}$ is constrained to be diagonal (i.e. $\operatorname{diag}(\boldsymbol{w})$), and parameters can be transformed to that form with a small manipulation.

**Claim 4.3.** *Given nonnegative matrices $\boldsymbol{B} \in \mathbb{R}_+^{n \times k_B}$ and $\boldsymbol{C} \in \mathbb{R}_+^{n \times k_C}$, letting $k = k_B + k_C$, there exist a matrix $\boldsymbol{V} \in [0,1]^{n \times k}$ and a diagonal matrix $\boldsymbol{W} \in \mathbb{R}^{k \times k}$ such that $\boldsymbol{VWV}^\top = \boldsymbol{BB}^\top - \boldsymbol{CC}^\top$.*

*Proof.* Let $\boldsymbol{m}_B$ and $\boldsymbol{m}_C$ be the vectors containing the maximums of each column of $\boldsymbol{B}$ and $\boldsymbol{C}$, respectively. The equality and the constraints on $\boldsymbol{V}$ and $\boldsymbol{W}$ are satisfied by setting

$$\boldsymbol{V} = \left(\boldsymbol{B} \times \operatorname{diag}\left(\boldsymbol{m}_B^{-1}\right); \quad \boldsymbol{C} \times \operatorname{diag}\left(\boldsymbol{m}_C^{-1}\right)\right)$$
$$\boldsymbol{W} = \operatorname{diag}\left(\left(+\boldsymbol{m}_B^2; \quad -\boldsymbol{m}_C^2\right)\right).$$

∎

---

**Algorithm 1** Fitting the Unconstrained LPCA Model

**input** adjacency matrix $\boldsymbol{A} \in \{0,1\}^{n \times n}$, rank $k < n$, regularization weight $\lambda \geq 0$, number of iters. $I$
**output** factors $\boldsymbol{X}, \boldsymbol{Y} \in \mathbb{R}^{n \times k}$ such that $\sigma(\boldsymbol{XY}^\top) \approx \boldsymbol{A}$
1: Initialize elements of $\boldsymbol{X}, \boldsymbol{Y} \in \mathbb{R}^{n \times k}$ randomly
2: **for** $i \leftarrow 1$ to $I$ **do**
3: $\quad \tilde{\boldsymbol{A}} \leftarrow \sigma(\boldsymbol{XY}^\top)$ $\hfill \triangleright$ reconstructed adjacency matrix
4: $\quad R \leftarrow -\sum\left(\boldsymbol{A}\log(\tilde{\boldsymbol{A}})\right) - \sum\left((1-\boldsymbol{A})\log(1-\tilde{\boldsymbol{A}})\right)$ $\hfill \triangleright$ cross-entropy loss
5: $\quad R \leftarrow R + \lambda\left(\|\boldsymbol{X}\|_F^2 + \|\boldsymbol{Y}\|_F^2\right)$ $\hfill \triangleright$ regularization loss
6: $\quad$ Calculate $\partial_{\boldsymbol{X},\boldsymbol{Y}} R$ via differentiation through Steps 3 to 5
7: $\quad$ Update $\boldsymbol{X}, \boldsymbol{Y}$ to minimize $R$ using $\partial_{\boldsymbol{X},\boldsymbol{Y}} R$
8: **end for**
9: **return** $\boldsymbol{X}, \boldsymbol{Y}$

---

**Implementation details** Our implementation uses PyTorch (Paszke et al., 2019) for automatic differentiation and minimizes the loss using the SciPy (Jones et al., 2001) implementation of the L-BFGS (Liu & Nocedal, 1989; Zhu et al., 1997) algorithm with default hyperparameters and up to a maximum of 200 iterations for both stages of optimization. We set the magnitude of the regularization to 10 times the mean entry value of the factor matrices. We include code in the form of a Jupyter notebook (Pérez & Granger, 2007) demo in the supplemental material.

---

**Algorithm 2** Initializing the Constrained Model from LPCA Logits

---

**input** logit factors $X, Y \in \mathbb{R}^{n \times k}$
**output** $B, C \in [0, \infty)^{n \times 3k}$ such that $BB^\top - CC^\top \approx \frac{1}{2}\left(XY^\top + YX^\top\right)$

1: Set $Q \in \mathbb{R}^{n \times k}$ and $\lambda \in \mathbb{R}^k$ by truncated eigendecomposition
   such that $Q \times \mathrm{diag}(\lambda) \times Q^\top \approx \frac{1}{2}(XY^\top + YX^\top)$
2: $B^* \leftarrow Q^+ \times \mathrm{diag}(\sqrt{+\lambda^+})$, where $\lambda^+, Q^+$ are the positive eigenvalues/vectors
3: $C^* \leftarrow Q^- \times \mathrm{diag}(\sqrt{-\lambda^-})$, where $\lambda^-, Q^-$ are the negative eigenvalues/vectors
4: $B \leftarrow \left(\sqrt{2}\phi(B^*); \quad \sqrt{2}\phi(-B^*); \quad |C^*|\right)$    ▷ $\phi$ and $|\cdot|$ are entrywise ReLU and absolute value
5: $C \leftarrow \left(\sqrt{2}\phi(C^*); \quad \sqrt{2}\phi(-C^*); \quad |B^*|\right)$
6: **return** $B, C$

---

## 5 THEORETICAL RESULTS

SYMNMF and BIGCLAM, among other models for undirected graph, assume network homophily, which precludes low-rank representation of networks with heterophily. We first show that our model is highly expressive in that it can capture arbitrary homophilous and heterophilous structure: using a result from Chanpuriya et al. (2020), we show that our model can exactly reconstruct a graph using a number of communities that is linear in the maximum degree of the graph.

**Lemma 5.1** (Exact LPCA Embeddings for Bounded-Degree Graphs, Chanpuriya et al. (2020))**.** *Let $A \in \{0, 1\}^{n \times n}$ be the adjacency matrix of a graph $G$ with maximum degree $c$. Then there exist matrices $X, Y \in \mathbb{R}^{n \times (2c+1)}$ such that $(XY^\top)_{ij} > 0$ if $A_{ij} = 1$ and $(XY^\top)_{ij} < 0$ if $A_{ij} = 0$.*

**Theorem 5.2** (Interpretable Exact Reconstruction for Bounded-Degree Graphs)**.** *Let $A \in \{0, 1\}^{n \times n}$ be the adjacency matrix of a graph $G$ with maximum degree $c$. Let $k = 12c + 6$. For any $\epsilon > 0$, there exist $mV \in [0, 1]^{n \times k}$ and diagonal $W \in \mathbb{R}^{k \times k}$ such that $\left\|\sigma(VWV^\top) - A\right\|_F < \epsilon$.*

*Proof.* Lemma 5.1 guarantees the existence of matrices $X, Y \in \mathbb{R}^{n \times (2c+1)}$ such that $(XY^\top)_{ij} > 0$ if $A_{ij} = 1$ and $(XY^\top)_{ij} < 0$ if $A_{ij} = 0$. Let $L = \frac{1}{2}(XY^\top + YX^\top)$, the symmetrization of $XY^\top$. Since $A$ is symmetric, it still holds that $L_{ij} > 0$ if $A_{ij} = 1$ and $L_{ij} < 0$ if $A_{ij} = 0$; further, as the sum of two rank-$(2c+1)$ matrices, the rank of $L$ is at most $2 \cdot (2c+1)$. Finally, by Claim 4.2 and Theorem 4.3, with $k = 3 \cdot 2 \cdot (2c+1) = 12c + 6$, there exist matrices $V \in [0, 1]^{n \times k}$ and diagonal $W \in \mathbb{R}^{k \times k}$ such that $VWV^\top = L$, meaning still $(VWV^\top)_{ij} > 0$ if $A_{ij} = 1$ and $(VWV^\top)_{ij} < 0$ if $A_{ij} = 0$. Since $\lim_{z \to -\infty} \sigma(z) = 0$ and $\lim_{z \to +\infty} \sigma(z) = 1$, it follows that

$$\lim_{s \to \infty} \sigma\left(V(sW)V^\top\right) = \lim_{s \to \infty} \sigma\left(sVWV^\top\right) = A,$$

that is, $W$ can be scaled larger to match $A$ arbitrarily closely. ∎

The above bound on the number of communities $k$ required for exact representation can be very loose. We additionally show that our model can exactly represent a natural family of graphs which exhibits both homophily and heterophily with small $k$. The family of graphs is defined below; roughly speaking, nodes in such graphs share an edge iff they coparticipate in some number of homophilous communities and don't coparticipate in a number of heterophilous communities. For example, the motivating graph described in Section 1 would be an instance of such a graph if there exists an edge between two nodes iff the two members are from the same location and have different roles (i.e., one is a recruiter and the other is a non-recruiter).

**Theorem 5.3.** *Suppose there is an undirected, unweighted graph on $n$ nodes with adjacency matrix $A \in \{0, 1\}^{n \times n}$ whose edges are determined by an overlapping clustering and a "thresholding" integer $t \in \mathbb{Z}$ in the following way: for each vertex $i$, there are two binary vectors $b_i \in \{0, 1\}^{k_b}$ and $c_i \in \{0, 1\}^{k_c}$, and there is an edge between vertices $i$ and $j$ iff $b_i \cdot b_j - c_i \cdot c_j \geq t$. Then, for any $\epsilon > 0$, there exist $V \in [0, 1]^{n \times (k+1)}$ and diagonal $W \in \mathbb{R}^{(k+1) \times (k+1)}$ such that $\left\|\sigma(VWV^\top) - A\right\|_F < \epsilon$.*

*Proof.* Let the rows of $B \in \{0, 1\}^{n \times k_b}$ and $C \in \{0, 1\}^{n \times k_c}$ contain the vectors $b$ and $c$ of all nodes. By Claim 4.3, we can find $V^* \in [0, 1]^{n \times k}$ and diagonal $W^* \in \mathbb{R}^{k \times k}$ such that $V^*W^*V^{*\top} =$

Table 1: Datasets used in our experiments. As in Sun et al. (2019), for YOUTUBE and AMAZON, we take only nodes which participate in at least one of the largest 5 ground-truth communities.

| Name | Reference | Nodes | Edges | Labels |
|------|-----------|-------|-------|--------|
| BLOG | Tang & Liu (2009) | 10,312 | 333,983 | 39 |
| YOUTUBE | Yang & Leskovec (2015) | 5,346 | 24,121 | 5 |
| POS | Mahoney | 4,777 | 92,406 | 40 |
| PPI | Breitkreutz et al. (2007) | 3,852 | 76,546 | 50 |
| AMAZON | Yang & Leskovec (2015) | 794 | 2,109 | 5 |

$BB^\top - CC^\top$. Now let

$$V = (V^* \quad 1) \qquad W = \begin{pmatrix} W^* & 0 \\ 0 & \frac{1}{2} - t \end{pmatrix}.$$

Then $(VWV^\top)_{ij} = b_i \cdot b_j - c_i \cdot c_j + \frac{1}{2} - t$. Hence $(VWV^\top)_{ij} > 0$ iff $b_i \cdot b_j - c_i \cdot c_j > t - \frac{1}{2}$, which is true iff $A_{ij} = 1$ by the assumption on the graph. Similarly, $(VWV^\top)_{ij} < 0$ iff $A_{ij} = 0$. It follows that

$$\lim_{s \to \infty} \sigma\left(V(sW)V^\top\right) = \lim_{s \to \infty} \sigma\left(sVWV^\top\right) = A.$$

∎

## 6 EXPERIMENTS

### 6.1 EXPRESSIVENESS

We investigate the expressiveness of our generative model, that is, the fidelity with which it can reproduce an input network. In Section 1, we used a simple synthetic network to show that our model is able to represent heterophilous structures in addition to homophilous structure. We now evaluate the expressiveness of our model on a benchmark of real-world networks, summarized in Table 1. As with the synthetic graph, we fix the number of communities or singular vectors, fit the model, then evaluate several types of reconstruction error. In Figure 3, we compare the results of our model with those of SVD, BIGCLAM (Yang & Leskovec, 2013), and SYMNMF (Kuang et al., 2015). The BIGCLAM model is discussed in detail in Section 3. SYMNMF simply factors the adjacency matrix as $A \approx HH^\top$, where $H \in \mathbb{R}_+^{n \times k}$; note that, like SVD, SYMNMF does not necessarily output a matrix whose entries are probabilities (i.e., bounded in $[0, 1]$), and hence it is not a graph generative model like ours and BIGCLAM.

For each method, we fix the number of communities or singular vectors at the number of ground-truth communities of the network. For a fair comparison with SVD, we do not regularize the training of the other methods. Our method consistently has the lowest reconstruction error, both in terms of Frobenius error and entrywise cross-entropy (Equation 3).

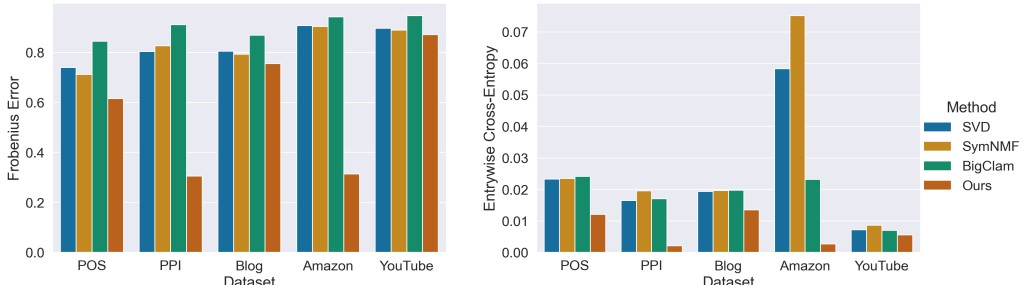

Figure 3: Error when reconstructing real-world graphs with SYMNMF, SVD, BIGCLAM, and our model. Frobenius error is normalized as in Figure 2; cross-entropy is normalized by the number of entries of the matrix ($n^2$).

## 6.2 SIMILARITY TO GROUND-TRUTH CLUSTERS

As a way of assessing the interpretability of the clusters generated by our method, we evaluate the similarity of the clusters to ground-truth communities, and we compare with results from other overlapping clustering algorithms. For all methods, we set the number of communities to be detected as the number of ground-truth communities. We report F1-Score as computed in Yang & Leskovec (2013). See Figure 4. The performance of our method is competitive with SYMNMF, BIGCLAM, and vGraph (Sun et al., 2019).

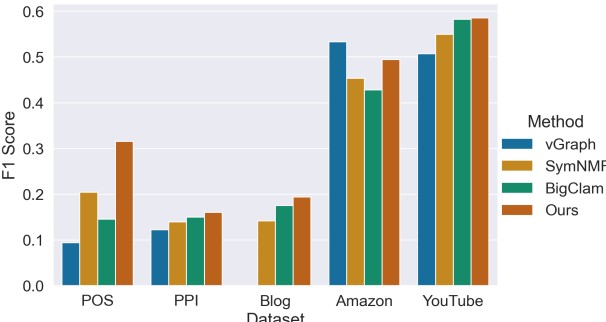

Figure 4: Similarity of recovered communities to ground-truth communities of real-world datasets. We were unable to run the authors' implementation of VGRAPH on BLOG with 16 GB of memory.

## 6.3 LINK PREDICTION

We assess the predictive power of our generative model via the link prediction task on real-world networks. As discussed in Section 2, the link probabilities output by our model are interpretable in terms of a clustering of nodes that it generates; we compare results with our method to those of other models which permit similar interpretation, namely BIGCLAM and SYMNMF. We randomly select 10% of node pairs to hold out (i.e. 10% of entries of the adjacency matrix), fit the models on the remaining 90%, then use the trained models to predict whether there are links between node pairs in the held out 10%. As a baseline for comparison, we also show results for randomly predicting link or no link with equal probability. See Figure 5 for the results. The performance of our method is competitive with or exceeds that of the other methods in terms of F1 Score.

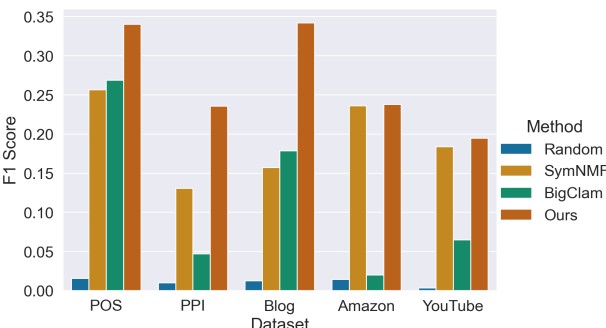

Figure 5: Accuracy of link prediction on real-world datasets.

## 7 CONCLUSION

We introduce an interpretable, edge-independent graph generative model that is highly expressive at representing both heterophily and overlapping communities. Our experimental results show its effectiveness on many important tasks. Further, our theoretical results demonstrate the expressiveness of our model in its ability to exactly reconstruct a graph using a number of clusters that is linear in the maximum degree, along with its ability to capture both heterophily and homophily in the data. In general, a deeper understanding of the expressiveness of both nonnegative and arbitrary low-rank logit models for graphs, as well as convergence properties of training algorithms, is an interesting direction for future research.

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
