# OpenReview forum: "An Interpretable Graph Generative Model with Heterophily"
_ICLR.cc/2022/Conference — ICLR 2022 Submitted_

### Official Review · Reviewer_gNYK · 2021-11-01

**Correctness:** 3
**Technical Novelty And Significance:** 3
**Empirical Novelty And Significance:** 2
**Recommendation:** 5
**Confidence:** 4

**Main Review:**

The model is a combination of three distinct features: a soft assignment matrix $V$, the use of a non-linearity $\sigma$, and finally a link matrix $W$ with arbitrary weights allowing for the modelling of heterophily. Many combinations of those three features are present in the literature (with Millet et al. being the closest one), but to the best of my knowledge this is the first general model that includes all three.

The algorithm proposed is technically sound, although the justification for steps 1 and 2 is tenuous: is it useful apart from selecting the right values for $k_B$ and $k_C$ ? This is an especially interesting question since step 2 involves the eigendecomposition of an $n \times n$ matrix, which implies an important computational cost. It is also further exacerbated by the fact that the supplementary material only implements step 3 of the algorithm, with $k_B$ and $k_C$ set to appropriate "magic" values and $B, C$ randomly initialized.

I find the section on theoretical results to be kind of a mixed bag. On the one hand, Theorem 5.3 is an interesting result, as it shows that the proposed algorithm covers a larger selection of mixed-membership models, including thresholding the matrix $\bar A$ instead of drawing Bernoulli variables. On the other hand, Theorem 5.2 seems to me more of a negative result: if any adjacency matrix can be simulated in the form of equation (1), even if there is no community structure, this casts a doubt on the interpretability of the algorithm output.

Finally, the experimental results are quite convincing, although they lack computation time analysis to really allow comparing algorithms together. It should maybe be noted that the results of Figure 1 are simply a consequence of Theorem 5.2, and the real interest of those experiments is to show that this approximation still captures the group assignments.

Overall, this article glossed quite quickly on a major interpretability problem: the model in equation (1) is invariant with respect to a lot of transformations (rotations, scalings...) the the matrices $V$ and $W$; this is used in the proof of Claim 4.3 and Theorems 5.2 and 5.3. I am unsure how all those transformations affect the community memberships, and whether the simpler model with $W$ diagonal has the same clustering effectiveness than the full model. Of course, this does not affect link prediction, but the main selling point of this article is its connection to clustering (else is it similar to e.g. Peysakhovich & Bottou) and it falls short on this part.

I am of course willing to revise my judgement if there is a simple resolution to this problem that I've missed and that can be added easily to the paper.


**Summary Of The Paper:**

This article deals with the task of community detection in the case of mixed membership networks, where a node can have multiple comunity assignments at once. It has been notice that in real-world networks, community overlaps tend to be much denser than the rest of the graph, which is a challenge to usual models (such as the stochastic block model).

The authors introduce a new mixed-membership graph model, where the expected adjacency matrix has the form

$$ \bar A = \sigma( V W V^\top), $$

with $\sigma$ the logistic function. In this model, the matrix $V$ represents the soft community assignments, and $W$ the inter-community similarities. The negative entries of $W$ allow this model to represent the heterophily of the network, wherein vertices are less likely to be connected when they belong to the same community.

They provide a learning algorithm to approximate $V$ and $W$ given the adjacency matrix $A$ of a graph, and show some existence results about such approximations.

In a second part, the authors provide several numerical experiments on real-world datasets, studying both the reconstruction of the matrix $\bar A$ and of the community membership matrix $V$.  They show that, in the chosen datasets, their algorithm outperforms or equals several others in the litterature.



**Summary Of The Review:**

This paper introduces a general model for mixed-membership community detection, along with an approximation algorithm to recover its parameters. This algorithms shows promising results on real-world datasets; however, the model has some interpretability issues that undermine its main selling point, and reduce its novelty compraed to the existing literature.

---

> ### Author Response · Authors · 2021-11-17
> **Response to Reviewer gNYK**
>
> Thank you very much for the thorough review. We appreciate the points raised, which we address below.
>
> Concerning the need for the first two training stages, indeed, one of the benefits of these stages is to automatically set the split between the number of homophilous/heterophilous communities $k_B$/$k_C$. Additionally, the computation in stage one is not wasted, since the arbitrary $X,Y$ are processed into nonnegative $B,C$ with approximately the same represented matrix; for a fair comparison, we provide each algorithm that we compare with the same total number of optimization steps, and we split these steps evenly between stages one and three for our algorithm. Further, the computational cost of stage two is not as great as a general eigendecomposition of an $n \times n$ matrix, since we seek only the top few ($O(k)$) eigenvectors, and the matrix we decompose is known in a rank-$O(k)$ factored form (so that matrix-vector multiplications are still fast).
> In experiments on synthetic graphs besides the one in Figure 1, we sometimes found that the community assignments with this multi-stage process were more consistent, so optimization through the first two stages may be better-conditioned; however, as shown in the demo in the supplemental material, it is often possible to simply fit the model directly.
>
> Concerning potential drawbacks of Theorem 5.2 and the motivating example, Theorem 5.2 states that the model can exactly represent graphs using a number of communities which is linear in the max degree of the graph. The significance of this result is mainly to contrast with positive semi-definite factorizations like BigClam, which cannot represent heterophily. The number of communities we fit is generally well below the bound of Theorem 5.2, and the use of regularization when training avoids exactly fitting the “observed” graph and rather fitting to the latent structure. The motivating graph from Figure 1 is an example of this: though it is fit to a sampled adjacency matrix, the reconstruction is qualitatively close to the expected adjacency matrix that generated the sample. Also on Figure 1, we find that the communities recovered by our model resemble noisy binary encodings of the ground-truth ones used to generate the network; this may be of interest to include.
>
> Concerning interpretability and the invariance of the model with respect to transformations, while the model is indeed invariant to certain rotations (e.g., permutations of the $k$ communities), it is not generally invariant to rotations due to the nonnegativity constraint.
> As for scaling transformations, indeed, it is possible to scale the assignments of all nodes to a given community up or down by a factor, and scale the corresponding entry in the community affinity matrix $W$ down or up, without changing the resulting model. We resolve this invariance by setting the maximum assignment for each community to be 1. As we noted above, our model produces quite reasonable community assignments on the motivating synthetic graph. As for real-world graphs, as we discuss in Section 6.2, when thresholding the soft community assignments to hard ones, our model’s assignments perform competitively or favorably relative to others in terms of matching ground-truth communities.
> Further, we believe that interpretability of the link predictions benefits from restricting $W$ to be diagonal: with this restriction, the logit for each edge prediction is a sum of only $k$ terms, as opposed to the $k^2$ terms that would result from an arbitrary $W$.

---

### Official Review · Reviewer_Fdty · 2021-11-02

**Correctness:** 3
**Technical Novelty And Significance:** 2
**Empirical Novelty And Significance:** 2
**Recommendation:** 3
**Confidence:** 4

**Main Review:**

Strengths :-
1) The paper discusses an important problem of modeling heterophily and overlapping communities in graph based generative models. Subsequently proposes an interesting approach to solving the same.
2) Via usage of non-negative matrix factorization (NMF), the approach in the paper can supposedly output link probabilities which are interpretable in terms of the detected communities.

Weaknesses :-
1) The approach proposed in the paper lacks motivation. Specifically, i) why did the authors choose to solve the problem using the three stage process, ii) what does each stage achieve, iii) can the approach be simplified from its complicated current form ?
2) I found the notation and descriptions in the paper to be confusing and wrong in several cases. Specifically I found the example demonstrated in Figures 1 and 2 to be confusing and not clear at all. In addition to this, it has typos - In figure 2, (right) and (left) in the captions are interchanged. Furthermore equation 1) is incorrect, i.e., how did the authors evaluate V.T * W * V when W is a k-by-k matrix and V is a n-by-k matrix is beyond me. I found parts of Section 4 to be unclear and confusing. To be more clear, the authors should have added more details about their training procedure and how they exactly achieve the different steps. I did not quite understand which part of their framework/approach or training procedure helps them capture/model the heterophily.
3) Building on weakness 1), I did not like the authors complicated three stage approximation procedure. Using NMF on top of that, builds in more error in the expressions. I am curious as to how much approximation error they incorporate during the entire process. Additionally I am not sure if one requires non-negativity of embeddings for interpretability.
4) There is no discussion of time/space complexity of their approach. I would like to see this detail in the paper.
5) The experiments are limited and use small datasets. I would have liked to see more analysis and discussion of the results. Also the authors should have compared their results against other more recent methodologies to demonstrate the clear superiority of their approach which is lacking in the paper. Via using small datasets, it adds more doubts to 4) above. Will their approach work on larger datasets ?


**Summary Of The Paper:**

In this work, the authors claim to propose the first edge-independent graph based generative model that can capture heterophily and via producing non-negative embeddings, it allows link predictions to be interpreted in terms of communities. The authors demonstrate the efficacy of their approach via empirical results in multi-label clustering and link prediction based tasks.

**Summary Of The Review:**

The current version of the draft needs some work. The paper lacks motivation and novelty in terms of its approach. The paper also needs i) more empirical results both with larger datasets, ii)comparison with more recent state-of-the-art approaches and iii) some discussion/analysis. Many parts of the paper are unclear and missing important details.

---

> ### Author Response · Authors · 2021-11-17
> **Response to Reviewer Fdty**
>
> Thank you for the review. We appreciate the concerns raised, which we address below.
>
> Thank you for bringing these typos to our attention; indeed, in Equation 1, the transpose should be on the second $V$.
>
> Concerning the intuition behind multiple training stages and the need for them, the overall idea is that the first stage produces a good initialization of the parameters for the third stage, while the second stage translates these parameters into a suitable form (nonnegative and symmetric) for the third stage. As we discuss in the response to Reviewer TiK2, these steps don’t necessarily add much complexity relative to simply starting at the third stage with random initialization. One of the benefits of these stages is that it automatically sets the split between the number of homophilous/heterophilous communities $k_B$/$k_C$; additionally, the overall procedure may be empirically better-conditioned than just doing the third stage.
>
> Concerning approximation error and modeling heterophily, our ultimate goal is not necessarily to match the input graph and minimize error as much as possible. We minimize the cross-entropy error with the goal of producing interpretable link predictions as expressed in our model. Specifically, the minimization of the error with the $\sigma(B B^\top - C C^\top)$ decomposition, along with regularization of the factors $B$ and $C$, can be seen as producing an “explanation” of the presence and absence of links in terms of homophilous/heterophilous cluster assignments $B$/$C$. Thus, fitting $C$ to minimize cross-entropy of link predictions results in finding heterophilous communities.
>
> Concerning the need for nonnegativity of embeddings for interpretability, there exist low-rank logit factorizations of graphs without nonnegativity, like logistic PCA, but these are generally not seen as interpretable. Our model offers a clustering interpretation for link logits, and the idea of the level of affinity of a node to a cluster is generally seen as nonnegative. As we discuss in our response to Reviewer gNYK, the nonnegativity constraint also improves interpretability in that it reduces the invariance of the cluster assignments to transformations like rotations.
>
> Concerning time complexity, details of training, and large datasets, as we discussed in the response to Reviewer TiK2, we mainly intend this work as a concept demonstration of our factorization, and experiments on larger datasets are an interesting direction. Each optimization step of the algorithm as presented in the work is $O(k n^2)$, and we simply use automatic differentiation for the gradient calculation. As we note, our loss function generalizes straightforwardly to a stochastic algorithm, where each step of optimization can be just $O(k)$. We include code for such an SGD version in the supplemental material.

---

### Official Review · Reviewer_TiK2 · 2021-11-02

**Correctness:** 2
**Technical Novelty And Significance:** 3
**Empirical Novelty And Significance:** 2
**Recommendation:** 3
**Confidence:** 5

**Main Review:**

The paper has a strong motivation over the generative graph model problem (GGM). It covers its use, and it explains some of the problems, especially the difficulty to capture heterophily. The paper also explains a type of GGMs and most importantly, it develops the training algorithm and some interesting theoretical results, which seems correct. So far, the paper is making an important contribution.

Unfortunately, there are some problems that must be addressed before publication.

First, the related work does not cover generative graph models (GGM). Instead, it focuses on node clustering, modeling heterophily, overlapping clustering. There are several works about GGMs that must be included in this paper. Especially, GGMs that are able to replicate the network structure and other characteristics, such as: Chung-lu, mKPGM, Bter, and others.

Second, it is difficult to figure out the main contribution inside the proposed generative graph model. Equation 1 is just one of the many ways to define the generative graph model. Then, the main contribution seems to be the training algorithm. However, this is based on a previous theory (Chanpuriya et al. (2020) and others). So it seems that the main contribution is the second stage, but it is not clear.

Third, the paper should explain why this model is able to model heterophily. The training process minimizes the sum of binary cross-entropies of the link predictions over all pairs of nodes. So, why does this minimization can model heterophily? How are they linked?

Forth, the implementation of this model is quite complicated and details are avoided. In line 6 from algorithm 1, where it is just mentioned "Calculate ∂X,Y R via differentiation through Steps 3 to 5". Please give more details about this step.

Fifth, add a time complexity analysis of the model, including the training and sampling process. While most generative graph models are able to reduce the time of the sampling process, as it is, the time complexity of the proposition is O(N^2), which is prohibitive for medium-size networks.

Sixth, given the low Frobenius error, it seems that the model suffers from the degenerative problem. This means that the model will generate almost the same network every time that is used to generate a network.

However, the main problem is the evaluation of the model. The paper proposes a generative graph model (GGM) and the evaluation is based on node clustering. The main purpose of GGMs is to replicate the characteristics and structure of the network. This is not evaluated in this paper. To do this, you have to sample multiple networks from your model and figure out if they can actually replicate some global and local network characteristics from the original network, such as clustering coefficient, geodesic distance, and others. Most importantly, the paper does not evaluate if the model is able to replicate heterophily, one of their main contribution.

**Summary Of The Paper:**

The paper proposes an edge-independent graph generative model. Besides the model, the paper also proposes the training process of the model and some theoretical contributions. Unfortunately, the evaluation of the model is based on another problem, clustering. This makes the paper complicate to evaluate. While the main contributions could be important, its evaluation does not guarantee that is able to replicate real networks. Most importantly, given its current results, it must suffer from the degeneracy problem.

**Summary Of The Review:**

While all contributions are very interesting, the evaluation of the paper is focused on another completely different problem. So, it is impossible to evaluate if the graph generative model is able to replicate the global and local characteristics of real networks.

---

> ### Author Response · Authors · 2021-11-17
> **Response to Reviewer TiK2**
>
> Thank you for the review. We appreciate the points raised, which we address below.
>
> 6\) Concerning degeneracy of the model, when our model is trained with very low regularization, indeed, it can have low entropy and simply reproduce the input graph. By contrast, in the limit of very high regularization, it will output an Erdős–Rényi model with $p=0.5$. We generally train with moderate regularization, producing models between these extremes. We focus on the low regularization case only in the motivating example and Section 6.1, to show that our model at least has the capability to match an input graph at low dimensionality, which other models may lack (e.g. BigClam is incapable of fitting the heterophilous structure in the motivating graph even as dimensionality is raised).
>
> 1\) Concerning generative graph models and evaluation, we mainly focus on how our factorization produces an expected adjacency matrix, that is, a symmetric matrix of probabilities that each edge appears. This is in contrast to related factorizations which do not naturally output edge probabilities (e.g., factorizations which would require thresholding entries to be in [0,1]). While we do not focus on the reconstruction of graph statistics by sampled graphs, our model is closely related to logistic PCA and others based on low-rank logit factorizations (e.g. NetGAN without GAN), which have performed comparably to deep generative graph models on graph statistic reconstruction.
>
> 2 & 3\) The main contribution is the introduction of the $\sigma(V W V^\top)$ and $\sigma(B B^\top - C C^\top)$ factorizations for undirected graphs with the respective restrictions on the factors, including nonnegativity of $B$ and $C$ (as we discuss, the two factorizations are equivalent). The factorizations model link probabilities in terms of overlapping communities, both homophilous/heterophilous communities, which correspond to positive/negative values of $W$ or the matrices $B$/$C$ as discussed in Sections 2 and 4. Thus, fitting $C$ to minimize cross-entropy of link predictions results in finding heterophilous communities. Some prior works proposed factorizations which have nonnegativity, but are not capable of heterophily (e.g. BigClam); other factorizations, like that of Peysakhovich et al., are capable of heterophily, but do not explore nonnegativity of the factors, which enhances the interpretability. Ours does both, and with Theorem 5.2, we also show that the constraint of nonnegativity does not lose any modeling capability.
>
> 4 & 5\) We mainly intend this work as a concept demonstration of our factorization, and experiments on larger datasets are an interesting direction. Indeed, each optimization step of the algorithm as presented in the work is $O(k n^2)$; we simply use automatic differentiation for this gradient calculation. As we note, our loss function generalizes straightforwardly to a stochastic algorithm, where each step of optimization can be just $O(k)$. We include code for such an SGD version in the supplemental material.

---

### Decision · Program_Chairs · 2022-01-20

**Decision:**

Reject

**Comment:**

The paper proposes an edge-independent graph generative model that can capture heterophily. The authors propose a 3-stage process to obtain the node representations. The idea of factorization in the form of BB^T-CC^T is an interesting approach to model heterophily.

The paper can be improved in terms of writing to better motivate the need for a 3-stage algorithm and how these individual steps are related to the existing techniques in the literature. The authors should elaborate on the implications of the theorems and the concerns raised by the reviewers in the body of the paper.

The algorithm faces scalability challenges, which are not studied well in the experiments. The reviewers also have raised concerns about degeneracy in network reconstruction experiments. Overall, the paper needs further improvements for publication.